# The Profile of MicroRNA Expression in Bone Marrow in Non-Hodgkin’s Lymphomas

**DOI:** 10.3390/diagnostics12030629

**Published:** 2022-03-03

**Authors:** Yuliya A. Veryaskina, Sergei E. Titov, Igor B. Kovynev, Tatiana I. Pospelova, Igor F. Zhimulev

**Affiliations:** 1Laboratory of Gene Engineering, Institute of Cytology and Genetics, SB RAS, 630090 Novosibirsk, Russia; 2Laboratory of Molecular Genetics, Department of the Structure and Function of Chromosomes, Institute of Molecular and Cellular Biology, SB RAS, 630090 Novosibirsk, Russia; titovse78@gmail.com (S.E.T.); zhimulev@mcb.nsc.ru (I.F.Z.); 3AO Vector-Best, 630117 Novosibirsk, Russia; 4Department of Therapy, Hematology and Transfusiology, Novosibirsk State Medical University, 630091 Novosibirsk, Russia; kovin_gem@mail.ru (I.B.K.); depart04@mail.ru (T.I.P.)

**Keywords:** Non-Hodgkin’s lymphomas, acute leukemia, bone marrow, microRNA, let-7a, miRNA-124

## Abstract

Non-Hodgkin’s lymphomas (NHLs) are a heterogeneous group of malignant lymphomas that can occur in both lymph nodes and extranodal sites. Bone marrow (BM) is the most common site of extranodal involvement in NHL. The objective of this study is to determine the unique profile of miRNA expression in BM affected by NHL, with the possibility of a differential diagnosis of NHL from reactive BM changes and acute leukemia (AL). A total of 180 cytological samples were obtained by sternal puncture and aspiration biopsy of BM from the posterior iliac spine. All the cases were patients before treatment initiation. The study groups were NHL cases (*n* = 59) and AL cases (acute lymphoblastic leukemia (*n* = 25) and acute myeloid leukemia (*n* = 49)); the control group consisted of patients with non-cancerous blood diseases (NCBDs) (*n* = 48). We demonstrated that expression levels of miRNA-124, miRNA-221, and miRNA-15a are statistically significantly downregulated, while the expression level of let-7a is statistically significantly upregulated more than 2-fold in BM in NHL compared to those in AL and NCBD. ROC analysis revealed that let-7a/miRNA-124 is a highly sensitive and specific biomarker for a differential diagnosis of BM changes in NHL from those in AL and NCBD. Therefore, we conclude that analysis of miRNA expression levels may be a promising tool for early diagnosis of NHL.

## 1. Introduction

Non-Hodgkin’s lymphoma (NHL) is a lymphoid tissue neoplasm arising from B cell precursors, mature B cells, T cell precursors, and mature T and NK cells. NHLs are a heterogeneous group of lymphoproliferative malignancies that can occur both in lymph nodes and in extranodal sites [1].

Determination of the international prognostic index (IPI) is an important step in assessing the prognosis of disease and choosing therapeutic approaches [2]. One of the IPI variables is assessing the status of bone marrow hematopoiesis. Bone marrow (BM) is the most common site of extranodal involvement in malignant lymphoid neoplasms [3]. It should be noted that the BM infiltration pattern can be different [4]. BM lesions can be detected using non-invasive diagnostic methods such as MRI and PET-CT, but BM biopsy is also indicated if PET/CT is negative; in some clinical situations, BM biopsy can be the only technique providing data on the presence and degree of BM infiltration with tumor lymphoid cells [5,6,7]. Currently, aspiration cytology and trephine biopsy are believed to complement each other and increase the accuracy of NHL diagnosis [8,9].

An important aspect of BM examination in NHL is detection of leukemic manifestations of lymphoma in the BM aspirate. An analysis of leukemic BM lesions in a cohort of Korean patients with NHL showed that the leukemic phase is associated with a poor prognosis: a low rate of complete remissions, a reduced 5-year survival rate, and a high mortality rate [10].

Now, one of the diagnostic problems in assessing BM involvement in patients with malignant lymphomas is the difficulty in distinguishing between reactive follicular infiltrates of non-neoplastic immune cells and the nodular infiltrative impairment of bone marrow hematopoiesis by malignant lymphoma. The development of molecular genetic approaches for the differential diagnosis of these hematopoietic pathological conditions would improve the quality of the primary diagnosis of NHL. It should be noted that the presence of tumor lymphocytes in BM can reflect the presence of NHL without involvement of the lymph nodes and other organs, i.e., the preclinical stage of blood cancer. Therefore, new diagnostic approaches to the detection of early phases of BM involvement may facilitate the early diagnosis of NHL.

Both genetic and epigenetic regulatory mechanisms, including microRNAs (miRNAs), are involved in tumor progression of malignant lymphomas with impairment of normal hematopoiesis. MiRNAs are short non-coding RNAs, and their aberrant expression can promote the development of hematological tumors [11]. To date, a number of studies have been published, which analyze levels of miRNA expression in lymph node tissue in NHL [12,13,14]. It has been shown that miRNAs can be used as prognostic biomarkers in NHL [15,16]. In NHL, both the primary site of tumor localization and the BM condition can reflect the development of disease. Therefore, the profile of miRNA expression in BM in NHL may reflect the development of malignant lymphoma.

We want to analyze the miRNA expression profile of not an individual cell type, but rather a total transcriptome of a bone marrow sample. Our goal is to compare the total miRNA profile of all cells in the samples of bone marrow and extracellular fluid of different groups of patients: with and without lymphoma. We want to identify specific epigenetic abnormalities to significantly distinguish BM lesions between the lymphoma and comparison groups. The reason for difference between the groups may be the presence of single tumor cells in the BM of lymphoma patients that cannot be identified by routine methods.

The objective of this study is to identify a unique profile of miRNA expression in BM affected by NHL, with the possibility of a differential diagnosis of NHL from reactive BM changes and acute leukemia (AL).

## 2. Materials and Methods

### 2.1. Clinical Samples

A total of 180 cytological samples were obtained by sternal puncture and aspiration biopsy of BM from the posterior iliac spine. All the cases were patients before treatment initiation. The study groups were NHL cases (*n* = 59) (small lymphocytic lymphoma (SLL) (*n* = 8), mantle cell lymphoma (MCL) (*n* = 6), follicular lymphoma (FL) (*n* = 7), marginal zone lymphoma (MZL) (*n* = 11), diffuse large B-cell lymphoma (DLBCL) (*n* = 27)) and AL (acute lymphoblastic leukemia (ALL) (*n* = 25) and acute myeloid leukemia (AML) (*n* = 49)); the control group consisted of patients with non-cancerous blood diseases (NCBDs) (*n* = 48) (iron-deficiency anemia (*n* = 28), hemolytic anemia (*n* = 3), B12 deficiency anemia (*n* = 5), chronic disease anemia (*n* = 6), immune thrombocytopenia (*n* = 5), aplastic anemia (*n* = 1)). When recruiting patients, we avoided obvious leukemic forms of indolent lymphomas. The characteristics of the groups are shown in Appendix A. Cytological material was obtained in compliance with Russian laws and regulations, written informed consent was obtained from each patient, and all the data were depersonalized. This study was approved by the ethics committee of the Novosibirsk State Medical University.

### 2.2. Isolation of Total RNA

Nucleic acids were extracted from fine-needle aspiration cytological specimens as described previously [17]. Each dried cytological specimen was washed in a microcentrifuge tube with three 200 μL portions of guanidine lysis buffer. Samples were vigorously mixed and incubated in a thermal shaker at 65 °C for 15 min. Next, an equal volume of isopropanol was added. The solution was thoroughly mixed and kept at room temperature for 5 min. After centrifugation at 14,000× *g* for 10 min, the supernatant was decanted, and the pellet was washed with 500 μL of 70% ethanol and 300 μL of acetone. The resulting RNA was dissolved in 290 μL of deionized water.

### 2.3. Selection of miRNAs

MiRNAs were selected based on literature data. Experimental analysis involved 17 miRNAs: miR-16-5p, -155-5p, -124-3p, -221-3p, -181a-5p, -15a-5p, -30a-5p, -182-5p, -7-5p, -196b-5p, -20a-5p, -23b-3p, -26a-5p, -29b-3p, -10b-5p, -145-5p, and let-7a-5p [11,18,19,20,21,22]. The reference miRNA was the geometric mean of Ct values of three reference miRNAs: miR-378-3p, -191-5p, and -103a-3p, which were selected based on our original data and literature data [23]. To quantify miRNA, we followed the protocol which includes reverse transcription of mature miRNA using long stem-loop primer, which is followed by detection of cDNA via RT-qPCR [24]. The oligonucleotide sequences are given in Appendix A. All the oligonucleotides, including fluorescently labeled ones, were synthesized at Vector-Best (Novosibirsk, Russia). The oligonucleotides were selected using an online tool, PrimerQuest (https://eu.idtdna.com/ (accessed on 12 January 2022)). For each miRNA, several sets of oligonucleotides were chosen, from which those with the highest real-time PCR efficiency were selected. The PCR efficiency was assessed by constructing a standard curve for serial dilutions of synthetic miRNA analogs (Biosan, Novosibirsk, Russia) of a known concentration. Depending on the system, the E value varied from 91.5% to 99.8%. The geometric mean of threshold cycle (Ct) values of the three reference miRNAs were used for normalization, as proposed by Vandesompele [25].

### 2.4. Reverse Transcription

The reverse transcription reaction for cDNA synthesis was carried out in a volume of 30 µL. The reverse transcription reaction contained 3 μL of RNA preparation, RT buffer with RT primer (Vector-Best, Novosibirsk, Russia) and Total RNA concentration is in the range (67–105) ng/µL, OD 260/280 ≥ 1.9 and OD 260/230 ≥ 1.5. Reaction was incubated for 15 min at 16 °C and 15 min at 42 °C, which was followed by heat inactivation for 2 min at 95 °C. Three microliters of RT mix was used per one RT-qPCR reaction.

### 2.5. Real-Time PCR

MiRNA expression levels were measured by real-time PCR on a CFX96 amplifier (Bio-Rad Laboratories, Hercules, CA, USA).

Total volume of each reaction was 30 μL and encompassed 3 μL of cDNA, 1× PCR buffer (Vector-Best, Russia), 0.5 μM of each primer and 0.25 μM of dual-labeled probe. PCR cycling conditions were as follows: 2 min incubation at 50 °C, pre-denaturation step at 94 °C—2 min—followed by 50 cycles of denaturation (94 °C for 10 s), annealing and elongation (60 °C for 20 s).

### 2.6. Statistical Analysis

Statistical analysis was performed using Statistica v13.1. The analysis was performed with the Mann–Whitney U test. *p*-values < 0.05 were considered statistically significant. The Bonferroni correction was applied to correct for multiple testing. The diagnostic values were evaluated by receiver operating characteristic (ROC) curves analysis (IBM SPSS software platform).

### 2.7. Functional Analysis of miRNAs Using DIANA-miRPath

The pathway analysis was carried out by DIANA-miRPath v3.0 (Thessaly, Greece) using *p* < 0.05 as a significant threshold.

## 3. Results

### 3.1. The Profile of miRNA Expression in BM Tumor Specimens and NCBD

MiRNA expression levels were measured by RT-qPCR in the NHL, AL, and NCBD groups (Figure 1).

A comparative analysis of miRNA expression levels between the cancer and NCBD samples revealed that miR-124, miR-221, miR-181a, miR-196b, miR-26a, and miR-15a were downregulated, and let-7a was upregulated in NHL (*p* < 0.05); miR-26a and miR-29b were downregulated in AL (*p* < 0.05) (Table 1). A comparative analysis of miRNA expression levels between the NHL and AL samples demonstrated that miR-124, miR-221, miR-181a, miR-196b, miR-23b, and mir-15a were downregulated, and let-7a was upregulated in NHL (*p* < 0.05) (Table 1).

There is a statistically significant decrease in the expression levels of miR-181a, miR-196b, and miR-26a in NHL in comparison with NCBD; however, a comparison of expression levels of these miRNAs between AL and NCBD shows a trend towards a decrease in the expression levels. Therefore, we think that miR-181a, miR-196b, and miR-26a cannot be used to differentiate NHL from AL and NCBD. Meanwhile, the levels of miR-124, miR-221, and miR-15a expression were statistically significantly downregulated more than 2-fold (*p* < 0.05) in NHL compared to both AL and NCBD. Regarding expression levels of miR-124, miR-221, and miR-15a, there is a trend towards upregulation of the expression levels in AL compared to NCBD. The expression level of let-7a is significantly upregulated in the NHL group compared to NCBD and AL groups (*p* < 0.05); in this case, there is a downregulation of the expression level in the AL group compared to the NCBD group.

A comparative analysis of miRNA expression levels between different types of NHL and NCBD samples revealed that miR-124 was downregulated (*p* < 0.05) in MCL, FL, MZL, DLBCL and tends to decrease the level of expression in SLL; miR-221 was downregulated (*p* < 0.05) in MZL and DLBCL and tends to decrease the level of expression in SLL, MCL, FL; miR-15a was downregulated (*p* < 0.05) in MZL, FL and DLBCL and tends to decrease the level of expression in SLL and MCL; miR-196b was downregulated (*p* < 0.05) in SLL, MZL, FL and DLBCL and tends to decrease the level of expression in MCL; and let-7a was upregulated (*p* < 0.05) in MZL and DLBCL and tends to increase the level of expression in SLL, FL and MCL (Table 2).

A comparative analysis of miRNA expression levels between the different types of NHL and AL samples demonstrated that miR-124 was downregulated (*p* < 0.05) in MCL, FL, MZL, DLBCL and tends to decrease the level of expression in SLL; miR-221 was downregulated (*p* < 0.05) in MZL and DLBCL and tends to decrease the level of expression in SLL, MCL, FL; miR-196b was downregulated (*p* < 0.05) in MZL and DLBCL and tends to decrease the level of expression in SLL, MCL and FL; miR-15a was downregulated (*p* < 0.05) in MZL, FL and DLBCL and tends to decrease the level of expression in SLL and MCL, and let-7a was upregulated (*p* < 0.05) in FL, MZL and DLBCL and tends to increase the level of expression in SLL and MCL (Table 3).

We showed a significant decrease of the expression levels of miRNA-124, miRNA-221 and miRNA-15a and an increase of the expression level of let-7a in various subtypes of NHL in comparison with both AL and NCBD.

### 3.2. Diagnostic Significance

To investigate the possibility of using miRNA-124, miRNA-221, miRNA-15a, and let-7a as new potential biomarkers for a differential diagnosis of BM changes in NHL from those in AL and NCBD, we performed ROC analysis in two selected cohorts. The NHL group and the group containing BM samples from AL and NCBD patients were compared. Figure 2 shows ROC curves for miRNA-124, miRNA-221, miRNA-15a, and let-7a, and Table 4 presents AUC, sensitivity, and specificity values of the tested biomarkers. The results demonstrate that miR-124 and let-7a are highly sensitive and specific for the diagnosis of BM changes associated with NHL.

Next, we evaluated whether a complex biomarker, which is the ratio of let-7a and miRNA-124 expression levels, may have a higher AUC value than each of these biomarkers alone. Figure 2E shows the ROC curve for a let-7a/miRNA-124 combination, with the AUC value being 0.908. Therefore, the use of a complex biomarker increases the sensitivity and specificity for NHL diagnosis.

### 3.3. Bioinformatic Analysis of Pathways Involved in Cancer Processes

We conducted a bioinformatic analysis of cancer-associated pathways that are regulated by miRNA-124 and let-7a. DIANA-TarBase v8 revealed 36 enriched pathways with an adjusted *p* < 0.05. Next, we focused on 12 pathways directly associated with cancer. The results are summarized in Table 5.

## 4. Discussion

NHLs are known to be located both in the lymph nodes and in the extranodal sites [1]. Obviously, the search for biomarkers that can improve the quality of early diagnosis of these tumors is the priority of modern medicine. There is evidence that miRNAs can be transported in exosomes [26]. Therefore, the BM condition may reflect the presence of a local tumor. We demonstrated that there is a unique expression profile of miR-124, miR-221, miR-15a, and let-7a in NHL in contrast to AL and NCBD (*p* < 0.05). However, regarding the practical application of these biomarkers for diagnostic purposes, only miRNA-124 and let-7a are characterized by the suitable AUC, sensitivity, and specificity indicators. Next, we tested the characteristics of a complex biomarker that included the ratio expression levels of let-7a and miRNA-124. We found that let-7a/miRNA-124 is the best of the tested biomarkers for diagnosing BM changes associated with NHL.

We found significant downregulation of the miRNA-124 expression level, more than 5-fold, in NHL compared to that in NCBD (*p* < 0.05). MiRNA-124 has been shown to be a tumor suppressor, and a decrease in its expression level is typical of tumors of various localization, but there is no evidence of the role of miRNA-124 in the development of NHL [27]. There are a number of studies reflecting the involvement of miRNA-124 in hematopoiesis. Liu et al. showed that miRNA-124 regulates Tip110 that is involved in the differentiation of hematopoietic stem cells [22]. Obviously, the aberrant expression of miRNA-124 can promote the development of hematological tumors. Kim et al. demonstrated that miRNA-124 regulates glucocorticoid sensitivity through PDE4B inhibition in the treatment of hematological malignancies [28]. Later, Shim et al. showed that miRNA-124 is a tumor suppressor in diffuse large B-cell lymphoma (DLBCL), and this tumor is characterized by a decreased expression level of miRNA-124 [29]. Jeong et al. identify miRNA-124 as a negative regulator of MYC and BCL2 expression in B-cell lymphomas and the genetic inhibition of this miRNA enhanced the fitness of these tumors [30]. These data are consistent with our findings. Thus, a decrease in the miRNA-124 expression level in BM may be associated with a local NHL.

To date, a number of studies have revealed the potential of miRNA as a diagnostic biomarker in NHL, but the study material is usually the affected lymph nodes, and the objective is differential diagnosis among NHL subtypes [28,29,30,31,32,33,34,35]. Lenze et al. note that miR-23a, -26a, -29b, -30d, -146a, -146b-5p, -155, and -221 are differentially expressed in DLBCL compared to Burkitt’s lymphoma (BL) [36]. An important discovery is the identification of miRNA-26a as a biomarker capable of differentiating unclassifiable B-cell lymphomas with features between DLBCL and BL [37]. Lawri et al. note the possibility of using miR-223, 217, 222, 221, let-7i, and let-7b as markers in diagnosis of follicular lymphoma (FCL) [38].

Tan et al. showed that miR-155, miR-181b, miR-15a, miR-16, miR-15b, miR-34a, miR-9, miR-30, let-7a, miR-125b, miR-217, and miR-185 modulate the expression of genes involved in B-cell maturation [39]. Another study notes that let-7a down-regulates MYC and reverts MYC-induced growth in Burkitt lymphoma cells [40]. Zhang et al. found that Myc represses miR-15a/miR-16-1 expression through the recruitment of HDAC3 in mantle cells and other non-Hodgkin B-cell lymphomas [41]. In addition, Georgantas et al. showed that miRNA-221 is involved in the regulation of normal hematopoiesis [42]. Sole et al. note that miRNA-181a is involved in the regulation of the early development of B cells [43]. Therefore, aberrant expression of these miRNAs may promote tumor development.

In our study, we note a significant downregulation of the expression levels of miR-124, miR-221, miR-181a, miR-196b, miR-26a, and miR-15a and an upregulation of the expression levels of let-7a in BM in NHL compared to those in NCBD (*p* < 0.05). Taken together, our findings in the BM material and the results of previous studies on lymph nodes indicate the role of these miRNAs in the pathogenesis of NHL.

Using the DIANA miRPath v.3.0 software, we found a link between miRNA-124, let-7a and cancer-associated genetic pathways. These results suggest that aberrant miRNA-124 and let-7a expression may affect cellular processes associated with cancer, contributing to the initiation, development, and progression of neoplasms.

Morphological assessment of bone marrow biopsy specimens is the standard method for analyzing BM lesions in NHL. Flow cytometry is an additional approach for evaluating BM in NHL, especially when detecting a small number of lymphoma cells [44]. However, this method also has limited sensitivity. In addition, an analysis for mutations typical of certain NHL types can be used as a diagnostic tool. Unfortunately, no universal mutations typical of all NHL types have been found so far. Therefore, the absence of mutations does not mean the absence of tumor cells. Thus, the search for universal diagnostic biomarkers in BM in patients with NHL is an important task of modern oncohemotology.

Epigenetic alterations of gene expression are important in the development of cancer. There are literature data on the methylation status in lymphoma cell lines. Bethge et al. demonstrated that the promoters of DSP, FZD8, KCNH2, and PPP1R14A were methylated in B-cell lymphoma samples [45]. However, these data are not intended to be used as markers for the early diagnosis of NHL.

We have shown that the BM condition may reflect the presence of NHL, and let-7a/miRNA-124 may be used as a diagnostic biomarker. However, the development of a diagnostic panel consisting of several miRNAs is a more optimal diagnostic tool. Therefore, it is necessary to search for additional biomarkers that can improve the quality of NHL diagnosis. Even though let-7 has been demonstrated to have tumor suppressive effects in various cancer types, emerging data suggest that, counterintuitively, in some cases let-7 may act as an oncogene. The overexpression of let-7a in lung cancer cells results in the increased aggressiveness of cells, assessed via an anchorage independent assay and an increase in gene expression associated with cell proliferation, as well as a downregulation of genes associated with adhesion, relevant to tumor progression and metastasis [46]. The let-7 family has been shown to influence the pathogenesis of a variety of hematological malignancies through the changing expression of a number of oncogenic pathways, particularly those related with MYC and that might affect hematopoietic carcinogenesis through the modulation of inflammatory pathways [47].

By analyzing miRNA levels, the authors hoped that identification of a bone marrow miRNA profile specific for neoplastic lymphoid proliferation would be a more sensitive and specific technique for the early diagnosis of bone marrow lymphomas or evaluation of the degree of lymphoma remission in hematological practice when tumor elements are not detected in the bone marrow by flow immunocytofluorimetry and classical morphology methods.

The authors agree that the research methodology differs from the classical one. We have tried to avoid the common classification of lymphoma types and consider them as a disease of the hematopoietic stem cell, in particular an abnormal lymphopoiesis progenitor cell. On this basis, we were able to combine the group of lymphomas and compare it with the group of non-neoplastic hematologic diseases. The purpose of this approach is to search for unified miRNA markers of abnormal gene expression in lymphoid tumors. Identification of a lymphoid tumor-specific miRNA profile from a wide range of regulators would enable using this profile in the early diagnosis of bone marrow lymphomas and the depth of disease remission of disease remission after completion of the standard treatment.

We are sure that there are universal epigenetic mechanisms of leukemia progression, which are independent of the immunomorphological variant of the latter. The data we obtained give us hope for identifying the pro-leukemic miRNA profile to further determine molecular and genetic markers for early disease detection and diagnosis of relapse. We hope that these markers will have a significant advantage over the existing diagnostic and prognostic criteria in their informative value.

The limitation of this study is a small number of samples for different types of NHL.

## Figures and Tables

**Figure 1 diagnostics-12-00629-f001:**
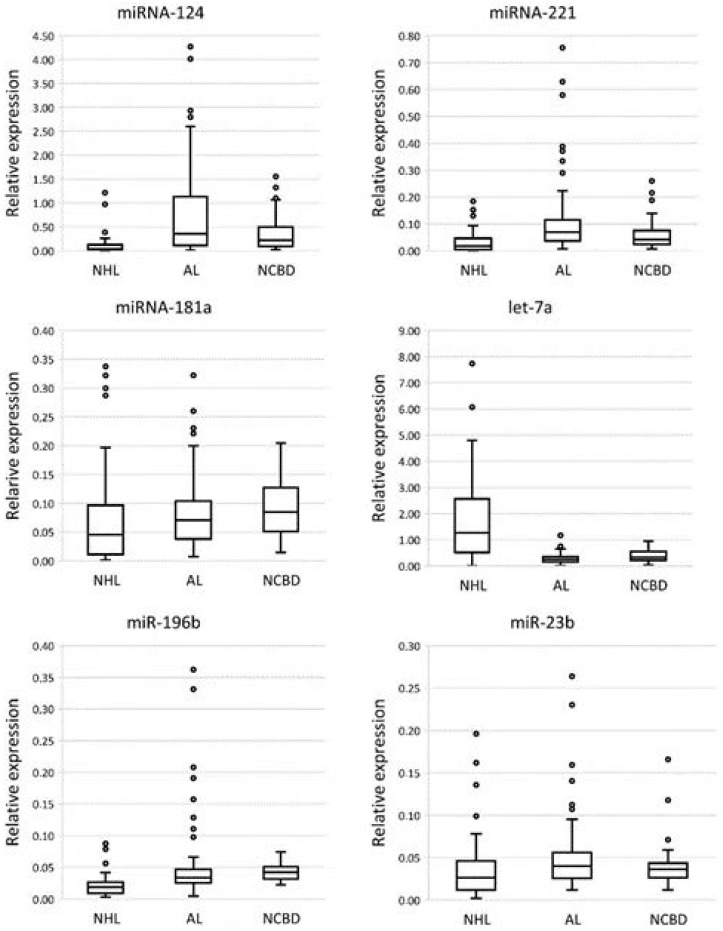
Comparative analysis of miRNA expression levels among non-Hodgkin’s lymphomas, acute leukemia, and non-cancerous blood diseases (NCBDs). The figure presents the median value, upper and lower quartiles, non-outlier range, and outliers appearing as circles.

**Figure 2 diagnostics-12-00629-f002:**
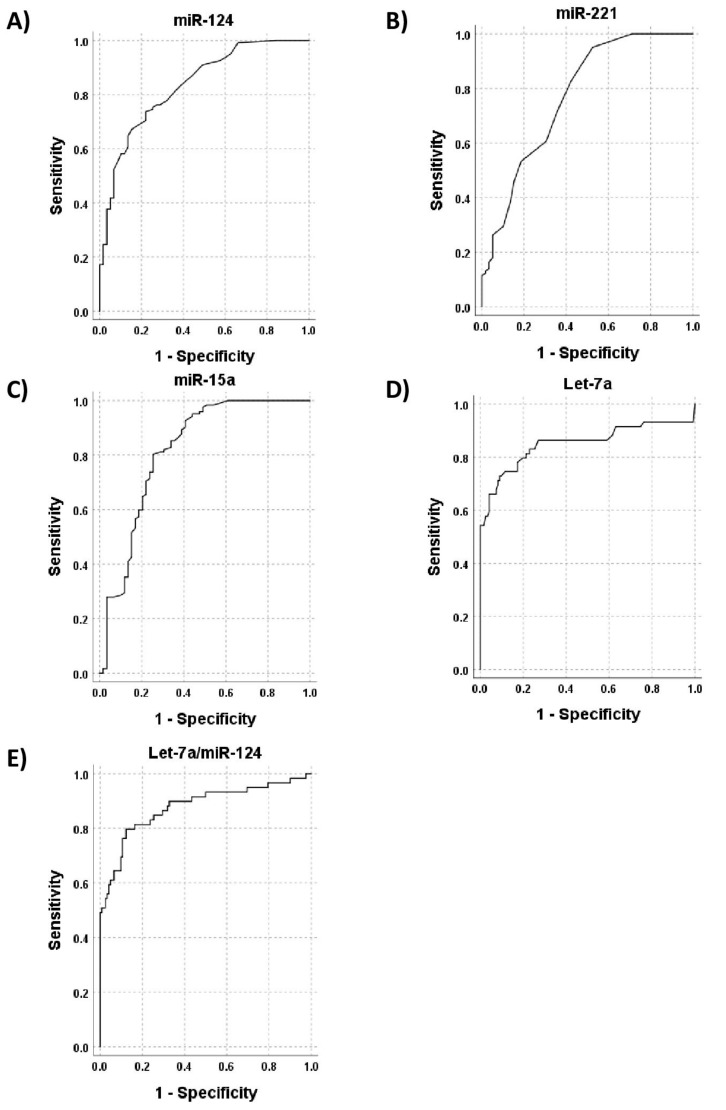
ROC Curves for miRNA-124, miRNA-221, miRNA-15a, and let-7a in differential diagnosis of BM changes in NHL from those in AL and NCBD. The ROC curve of miR-124 (**A**), miR-221 (**B**), miR-15a (**C**), let-7a (**D**), and let-7a/miR-124 combination (**E**).

**Table 1 diagnostics-12-00629-t001:** Comparative analysis of miRNA expression levels between tumor samples and NCBD.

	NHL vs. NCBD	Corrected *p*-Value	AL’s vs. NCBD	Corrected *p*-Value	NHL vs. ALs	Corrected *p*-Value
miR-16	1	NS	−1.23	NS	1.23	NS
miR-155	1.02	NS	1.04	NS	−1.03	NS
miR-124	−5.29	5 × 10^−8^	1.68	NS	−8.87	5 × 10^−10^
miR-221	−2.34	0.004	1.7	NS	−3.98	1 × 10^−10^
miR-181a	−1.63	0.007	−1.09	NS	−1.49	0.002
miR-30a	−1.17	NS	−1.22	NS	1.05	NS
miR-182	−1.42	NS	−1.04	NS	−1.37	NS
miR-7	−1.63	NS	−1.03	NS	−1.59	NS
let-7a	3.88	4 × 10^−7^	−1.3	NS	5.04	1 × 10^−13^
miR-196b	−2.22	1 × 10^−7^	−1.23	NS	−1.81	1 × 10^−7^
miR-20a	1	NS	−1.2	NS	1.21	NS
miR-23b	−1.37	NS	1.1	NS	−1.51	0.022
miR-26a	−2.36	NS	−1.59	0.006	−1.49	NS
miR-29b	−1.32	NS	−1.66	0.003	1.25	NS
miR-145	1.11	NS	−1.24	NS	1.37	NS
miR-15a	−4.25	3 × 10^−8^	1.01	NS	−4.3	1 × 10^−8^
miR-10b	−1.82	NS	1.08	NS	−1.98	NS

NS—not significant.

**Table 2 diagnostics-12-00629-t002:** Comparative analysis of miRNA expression levels between different types of NHL and NCBD.

	SLL vs. NCBD	Corrected *p*-Value	MCL vs. NCBD	Corrected *p*-Value	FL vs. NCBD	Corrected *p*-Value	MZL vs. NCBD	Corrected *p*-Value	DLBCL vs. NCBD	Corrected *p*-Value
miR-16	1.07	NS	1.39	NS	1.27	NS	−1.51	NS	1.22	NS
miR-155	6.04	NS	3.91	NS	−1.90	NS	−1.03	NS	−1.30	NS
miR-124	−1.66	NS	−5.29	0.006	−7.48	0.001	−4.41	0.016	−5.64	5 × 10^−7^
miR-221	−1.23	NS	−1.71	NS	−5.19	NS	−1.97	0.045	−5.66	2 × 10^−4^
miR-181a	1.02	NS	−1.30	NS	−2.42	NS	−1.48	NS	−2.10	2 × 10^−4^
miR-30a	2.00	NS	1.28	NS	−1.17	NS	−1.52	NS	−1.27	NS
miR-182	−1.04	NS	−1.05	NS	1.21	NS	−1.52	0.007	−1.15	NS
miR-7	1.27	NS	−1.31	NS	−1.14	NS	−3.49	NS	−2.09	0.008
let-7a	2.85	NS	6.46	NS	3.18	NS	2.73	1 × 10^−4^	6.29	1 × 10^−8^
miR-196b	−1.99	0.012	−2.09	NS	−1.82	1 × 10^−4^	−2.87	1 × 10^−8^	−2.22	2 × 10^−8^
miR-20a	1.48	NS	1.39	0.046	1.34	NS	−1.11	NS	−1.03	NS
miR-23b	1.28	NS	−1.10	NS	−2.02	NS	−1.69	NS	−1.43	0.015
miR-26a	−2.75	NS	−1.37	NS	−1.01	NS	1.00	NS	1.05	NS
miR-29b	2.07	NS	−1.40	NS	−1.28	NS	1.05	NS	−1.53	NS
miR-145	1.96	NS	−1.10	NS	1.46	NS	1.10	NS	1.22	NS
miR-15a	−2.51	NS	−1.25	NS	−2.45	0.014	−4.72	0.005	−5.78	5 × 10^−8^
miR-10b	2.65	N S	−2.20	N S	−2.32	N S	−2.33	NS	−1.65	NS

NS—not significant.

**Table 3 diagnostics-12-00629-t003:** Comparative analysis of miRNA expression levels between different types of NHL and AL.

	SLL vs. ALs	Corrected *p*-Value	MCL vs. ALs	Corrected *p*-Value	FL vs. ALs	Corrected *p*-Value	MZL vs. ALs	Corrected *p*-Value	DLBCL vs. ALs	Corrected *p*-Value
miR-16	1.34	NS	1.74	NS	1.59	NS	−1.21	NS	1.53	NS
miR-155	5.81	NS	3.75	NS	−1.98	NS	−1.08	NS	−1.36	NS
miR-124	−2.74	NS	−8.73	0.013	−1.98	0.001	−7.28	0.002	−9.31	1 × 10^−8^
miR-221	−2.13	NS	−2.96	NS	−12.35	NS	−3.41	2 × 10^−4^	−9.77	1 × 10^−7^
miR-181a	1.11	NS	−1.19	NS	−8.97	NS	−1.35	NS	−1.92	1 × 10−4
miR-30a	2.46	NS	1.58	NS	−2.22	NS	−1.23	NS	−1.03	NS
miR-182	1.02	NS	1.00	NS	1.27	NS	−1.44	NS	−1.09	NS
miR-7	1.31	NS	−1.28	NS	−1.10	NS	−3.39	NS	−2.03	0.036
let-7a	3.55	NS	8.05	NS	3.96	0.013	3.40	3 × 10^−8^	7.83	1 × 10^−14^
miR-196b	−1.63	NS	−1.71	NS	−1.49	NS	−2.35	5 × 10^−5^	−1.82	1 × 10^−4^
miR-20a	1.74	NS	1.63	0.009	1.57	NS	1.06	NS	1.14	NS
miR-23b	1.17	NS	−1.21	NS	−2.22	NS	−1.85	0.030	−1.57	0.003
miR-26a	−1.96	NS	−1.53	NS	1.11	NS	1.12	NS	1.17	NS
miR-29b	3.43	NS	1.18	NS	1.30	NS	1.74	NS	1.08	NS
miR-145	2.45	NS	1.14	NS	1.83	NS	1.37	NS	1.52	NS
miR-15a	−2.55	NS	−1.27	NS	−2.49	NS	−4.79	0.004	−5.87	2 × 10^−8^
miR-10b	2.39	NS	−2.44	NS	−2.58	N S	−2.58	NS	−1.83	NS

**Table 4 diagnostics-12-00629-t004:** AUC, sensitivity, and specificity values for miRNA-124, miRNA-221, miRNA-15a, and let-7a in differential diagnosis of BM changes in NHL from those in AL and NCBD.

	AUC	Sensitivity	Specificity
miR-124	0.835	74%	78%
miR-15a	0.816	81%	71%
miR-221	0.769	71%	64%
let-7a	0.852	79%	80%
let-7a/miR-124	0.908	91%	80%

**Table 5 diagnostics-12-00629-t005:** Cancer-associated pathways in which the miRNAs in question are involved. The list was generated by DIANA-mirPath v3.0.

	KEGG Pathway	Genes in the Pathway, Total	*p* Value
**miRNA-124**			
	**Cell biology**		
	Cell cycle (hsa04110)	26	5 × 10^−3^
	**Cancer-associated pathways**		
	Pathways in cancer (hsa05200)	80	5 × 10^−3^
	Proteoglycans in cancer (hsa05205)	41	1.25 × 10^−5^
	Chronic myeloid leukemia (hsa05220)	20	1 × 10^−2^
	Bladder cancer (hsa05219)	15	1 × 10^−3^
	Acute myeloid leukemia (hsa05221)	16	1 × 10^−2^
**Let-7a**			
	**Cell biology**		
	Cell cycle (hsa04110)	43	5.6 × 10^−9^
	**Cancer-associated pathways**		
	Pathways in cancer (hsa05200)	97	3 × 10^−3^
	Proteoglycans in cancer (hsa05205)	49	4 × 10^−4^
	Chronic myeloid leukemia (hsa05220)	20	2 × 10^−3^
	Bladder cancer (hsa05219)	13	3 × 10^−2^
	Transcriptional misregulation in cancer (hsa05202)	40	4 × 10^−3^
	**Signaling pathway**		
	Hippo signaling pathway (hsa04390)	42	2 × 10^−7^
	p53 signaling pathway (hsa04115)	23	1 × 10^−3^
	FoxO signaling pathway (hsa04068)	35	9 × 10^−3^
	MAPK signaling pathway (hsa04010)	51	7 × 10^−3^
	TGF-beta signaling pathway (hsa04350)	27	2 × 10^−3^

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
