# Peer review of "The Profile of MicroRNA Expression in Bone Marrow in Non-Hodgkin’s Lymphomas"

_diagnostics, 2022, doi:10.3390/diagnostics12030629_

Round 1
Reviewer 1 Report
GENERAL COMMENTS
The present study investigates the expression profile of selected miRNAs regarding their potential as biomarkers in NHLs. The selection of miRNAs was thorough and the findings could be helpful in this research field. However, there are several issues that should be addressed.
SPECIFIC COMMENTS
Major Comments
- The group of NHL was highly heterogenous including 45 patients with DLBCL, 6 MCL, 6 CLL and 2 FL. These disease subtypes have totally different biology and should be analyzed separately.
- The presence or absence of bone marrow involvement is not described. The bone marrow is always involved in CLL and frequently in MCL and FL but is uncommonly involved in DLBCL. The analysis should take into account this information.
- The authors should clarify the starting mass of RNA for the reverse transcription. Additionally, it is not clear, whether total RNA was reversely transcribed or only the part of miRNAs and the other small non-coding RNAs. If the second one is the case, the authors should explain in more detail the experimental procedure.
- It is not clear why the Authors decided to investigate the expression of miRNAs in NHL compared to acute leukemias.
- Since the biomarker utility of these miRNAs is evaluated, did the Authors consider conducting univariate and multivariate survival analysis (progression-free and overall survival)?
- In the section of Discussion, the Authors should analyze more in depth the potential functional role of these miRNAs and include a section with the limitations of the study.
Minor Comments
- Does table 3 include the merged results of the analysis of both miRNAs? If this is the case, I propose the results to be separated.
- The labels in Figure 2 should be enlarged.
- Please indicate the group that patients with NHL who were exactly 60 years old were included (≥60 or ≤60 years).
- The authors could include a Figure regarding the enriched pathways, so that the manuscript is more attractive to the readers. Additionally, they could include a Figure with the results of each miRNA, since their high number constitute the understanding of the results difficult to the reader.
Reviewer 2 Report
The paper entitled “The profile of MicroRNA Expression in Bone Marrow in Non-Hodgkin’s Lymphomas” by Veryaskina et al., is focused on the detection of new biomarkers for NHLs and the results highlighted four miRNA that can have an Important role in diagnostic of BM changes in NHLs distinguishing between NHL and AL. miRNA-124, miRNA-221 and miRNA-15a were detected as downregulated and let-7a/miRNA-124 upregulated in NHLs when comparing to AL, these four miRNAs being promising tools for NHL early diagnostic.
The paper is well written and in the study groups were included many samples from different patients, thus the information provided by the miRNA expression is relevant and has high scientific value.
The information presented in this paper is confirmed by the cited literature and the supplementary data provided with this manuscript is helpful, allowing the readers to fully understand the research topic.
The materials and methods are well described, with lot of details, and the statistical analysis follow other publications in this field.
The results section is well organized and the figures are clear and easy to understand. Moreover, the diagnostic significance was highlighted for the four miRNAs that were significantly up/down regulated and can play biomarker role.
Final comments:
Dear authors, the herein manuscript is well written, the presented scientific data is solid due to the high number of samples included in each experimental group and sustained by the literature.
Your study could rise interest of the readers and I recommend its publication.
Reviewer 3 Report
The objective of this study is to determine the unique profile of miRNA expression in BM affected by NHL, with the possibility of differential diagnosis of NHL from reactive BM changes and acute leukemia. Evaluation of down-regulation of miRNA-124, miRNA-221 and miRNA-15a expression level in contrast with upregulation of let-7a is clear and well described.
Althoug authors have tried to avoid the common classification of lymphoma types and consider them as a disease of the hematopoietic stem cell, analysis of the subgroups of lymphoma should be performed, in order to confirm the power of this analysis (at least in the major group of DLBCL).
